# Susceptibility mapping and zoning of highway landslide disasters in China

**Chao Yin[1,2], Haoran Li[3], Fa Che[4]\*, Ying Li[1], Zhinan Hu[5], Dong Liu[6]**

**1** School of Civil and Architecture Engineering, Shandong University of Technology, Zibo, China, **2** Key Laboratory of Roads and Railway Engineering Safety Control, Shijiazhuang Tiedao University, Ministry of Education, Shijiazhuang, China, **3** Urban Rail Construction Corporation, Zhongtian Construction Group Co., LTD, Hangzhou, China, **4** Zibo Transportation Service Center, Zibo, China, **5** State Key Laboratory of Mechanical Behavior and System Safety of Traffic Engineering Structures, Shijiazhuang Tiedao University, Shijiazhuang, China, **6** Laoling Branch of Dezhou Highway Development Center, Dezhou, China

\* zbsglgljyanghuke@zb.shandong.cn

**Data Availability Statement:** All relevant data are within the manuscript and its Supporting Information files.

**Funding:** This work was supported by the National Natural Science Foundation of China (Grant NO.

## Abstract

Prominent regional differentiations of highway landslide disasters (HLDs) bring great difficulties in highway planning, designing and disaster mitigation, therefore, a comprehensive understanding of HLDs from the spatial perspective is a basis for reducing damages. Statistical prediction methods and machine learning methods have some defects in landslide susceptibility mapping (LSM), meanwhile, hybrid methods have been developed by combining the statistical prediction methods with machine learning methods in recent years, and some of them were reported to perform better than conventional methods. In view of this, the principal component analysis (PCA) method was used to extract the susceptibility evaluation indexes of HLDs; the particle swarm optimization-support vector machine (PSO-SVM) model and genetic algorithm-support vector machine (GA-SVM) model were implemented to the susceptibility mapping and zoning of HLDs in China. The research results show that the accumulative contribution rate of the four principal components is 92.050%; evaluation results of the PSO-SVM model are better than those of the GA-SVM model; micro dangerous areas, moderate dangerous areas, severe dangerous areas and extreme dangerous areas account for 24.24%, 19.49%, 36.53% and 19.74% of the total areas of China; among the 1543 disaster points in the HLDs inventory, there are 134, 182, 421 and 806 located in the above areas respectively.

## 1 Introduction

Taking the highway slope as the disaster bearing body and the surrounding environment as the disaster pregnant environment, highway landslide disaster (HLD) is one of the main reasons for long-term highway interruption [1–3]. HLDs occur frequently in some areas of China, resulting in serious economic losses and casualties [4], for example, the volume of the K1428+800 landslide of G108 Shaanxi segment exceeded $1 \times 10^5$ m$^3$, resulting in highway interruption for more than 3 years [5]; the Jiuzhaigou Valley's 7.0-magnitude earthquake led to the formation of 1,594 landslides, covering a total volume of $11.52 \times 10^6$ m$^3$ [6, 7]. The prominent

51808327) and Natural Science Foundation of Shandong Province (Grant NO. ZR2019PEE016). The funders had no role in study design, data collection and analysis, decision to publish, or preparation of the manuscript. Urban Rail Construction Corporation, Zhongtian Construction Group Co., LTD is a commercial organization and provided support in the form of salary for Haoran Li. The specific roles of all authors were articulated in the "author contributions" section.

**Competing interests:** Haoran Li is affiliated with Urban Rail Construction Corporation, Zhongtian Construction Group Co., LTD, a commercial organization. This affiliation does not alter our adherence to PLOS ONE policies on sharing data and materials. There are are no patents, products in development or marketed products to declare.

regional differentiations of HLDs bring great difficulties in highway planning, designing and disaster mitigation, therefore, a comprehensive understanding of HLDs from the spatial perspective is a basis for reducing damages [1, 8–11]. Susceptibility mapping and zoning can reveal the spatial differentiations of HLDs and divide China into areas with different susceptible levels, thus to clarify the priorities and protection standards for different areas, and provide theoretical basis for macro mitigation policy formulation [3, 12].

Researches on landslide susceptibility mapping (LSM) in China mainly focused on the Wenchuan, Yushu and Ya'an earthquake areas, the Three Gorges Reservoir areas, the areas affected by typhoons and loess areas; researches abroad China mainly focused on the Medellin areas (Columbia), Kyushu areas (Japan) and some areas in Italy [13–15]. The modeling methods implemented to LSM mainly included the statistical prediction models, i.e., Logistic regression method (LR), decision tree method, analytical hierarchy process (AHP), deterministic coefficient method and multivariate adaptive regression spline model (MARSplines), and the machine learning models, i.e., artificial neural network (ANN), support vector machine (SVM), neuro-fuzzy technique, decision tree model and Bayesian network (BN), some scholars also conducted comparison researches on multiple modeling methods [11, 16–20]. Representative studies included: Wang et al. [21] used the LR, bivariate statistical analysis (BS) and MARSplines to create landslide susceptibility maps by comparing the past landslide distribution and conditioning factor thematic maps; Alireza et al. [22] proposed a novel hybrid model based on the step-wise weight evaluation ratio analysis (SWARA) method and adaptive neuro-fuzzy inference system (ANFIS) to evaluate landslide susceptible areas using geographical information system (GIS); Zhang et al. [23] used the information value model and LR to build the susceptibility evaluation systems based on the data of 655 landslides in the history of Wanzhou district (Chongqing); Sezer et al. [24] conducted landslide susceptibility evaluation by applying the methods of M-AHP and Mamdani type FIS by using the expert-based LSM module; Chen et al. [25] built a landslide susceptibility model using three well-known machine learning models namely the maximum entropy (MaxEnt), SVM and ANN, and accompanied by their ensembles (i.e., ANN-SVM, ANN-MaxEnt, ANN-MaxEnt-SVM and SVM-MaxEnt) in Wanyuan (China); Zhu et al. [26] developed and compared two presence-only methods including the one-class SVM and kernel density estimation (KDE), and two presence-absence methods including the ANN and two-class SVM to evaluate their respective performance in mapping landslide susceptibility; Chen et al. [11] assessed and compared four advanced machine learning techniques, namely the BN, radical basis function classifier (RBF), logistic model tree (LMT) and random forest (RF) models, for landslide susceptibility modeling in Chongren, China; Yang et al. [27] proposed a new LSM method based on the GeoDetector and spatial logistic regression model (SLR), of which, the GeoDetector was used to select condition factors based on the spatial distribution of landslides, SLR model was used to make full use of the structural and attribute information of spatial objects simultaneously in LSM.

There are still several defects of current researches on LSM: (1) Current researches generally focus on the view of physical geography, however, this unprofessional mapping cannot reflect on the mutual feedback mechanism between the occurrences of HLDs and their disaster pregnant environment, only provide indirect references for highway planning, designing and disaster mitigation [1, 3]; (2) SVM is one of the main modeling methods implemented to LSM, the critical factors affect its calculation efficiency are the optimization speeds of the penalty parameter $C$ and nuclear parameter $\sigma$, when the optimization scope is large, SVM often tends to consider the partial optimum as overall optimum, resulting in early maturity [28–30]. Hybrid methods have been developed by combining the statistical prediction methods with machine learning methods in recent years, some of them were reported to perform better than conventional methods [11]. In view of this, the principal component analysis (PCA) method was used

to extract the susceptibility evaluation indexes of HLDs; the particle swarm optimization-support vector machine (PSO-SVM) model and genetic algorithm-support vector machine (GA-SVM) model were implemented to the susceptibility mapping and zoning of HLDs in China, and the better evaluation model was determined through the areas under curve (AUC) method. The contents of this study include: (1) select the impact factors of HLDs; (2) compile the HLDs inventory; (3) extract the susceptibility evaluation indexes of HLDs by PCA method; (4) determine the better evaluation model by AUC method; (6) susceptibility mapping of HLDs; and (7) propose the susceptibility zoning scheme of HLDs in China, the flowchart of this study is showed in Fig 1.

## 2 Susceptibility evaluation indexes of HLDs

Susceptibility evaluation of HLDs is a kind of comprehensive evaluation, its object is to determine the intensity, frequency and density of HLDs according to the spatial distribution and combination characteristics of the disaster pregnant environment elements, i.e., analyzing the effects of the evaluation indexes and their combination characteristics on the occurring possibilities and scales of HLDs [31–34].

### 2.1 Impact factors of HLDs

Selecting impact factors is an important step in LSM because they may not be independent with each other, which can introduce noises and decrease the prediction capabilities of models

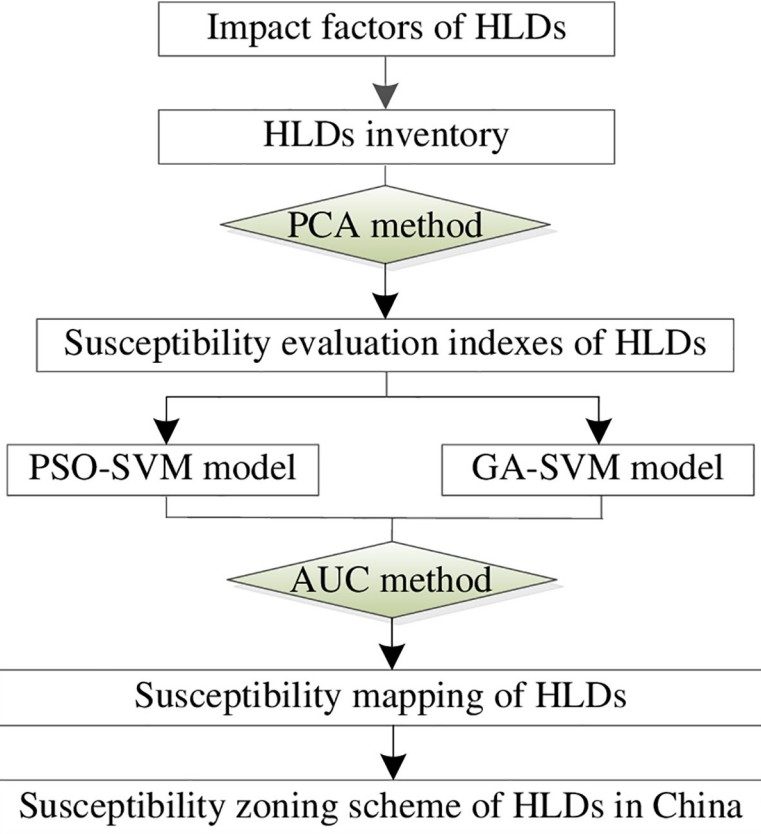

**Fig 1. Flowchart of this study.**

[10]. Impact factors of HLDs mainly include basic factors and inducing factors, i.e., slope, elevation, slope aspect, lithology, distance to faults, distance to rivers, normalized difference vegetation index (NDVI), land use, mean precipitation, profile curvature, stream power index (*SPI*) and topographic wetness index (*TWI*) [1, 3, 18, 33].

1. Slope and elevation directly determine the stress distribution of a highway slope, larger slope and elevation will lead to higher potential energy, so that weak structural plane will be exposed easily and the highway slope will suffer from instability [13, 35].

2. Slope aspect has important effects on the distribution of solar radiation and formation of regional microclimate, and also affects the growth of vegetation to a certain extent, which is one of the commonly used impact factors of LSM [35, 36].

3. Lithology is an important component in the sliding mechanism process and material basis to form HLDs, and has been widely used for modeling landslide susceptibility in previous studies [14, 37].

4. Faults are usually related to earthquakes and act as the main control on the weak boundary controlling the deformation and failure mode of a highway slope, the compressive fault also generates a large number of secondary structural planes in the rock mass within the affected areas [14, 38, 39].

5. Rivers can provide wet and saturated water of the sliding areas, which may reduce the shear strength of the soil and weak layer, and reduce the stability of a highway slope, so distance to rivers is usually considered as an important impact factor of LSM [20, 40].

6. NDVI is used to quantify the vegetation density, the areas with low NDVI values are featured with bare rock and soil, and bad water and soil conservation capacity, resulting in formulations of HLDs easily [41, 42].

7. Land use is an important landslide-related factor because it affects the formulations of HLDs due to human intervention, land use patterns consist of bareland, cropland, forest, grassland, residential land, wetland and waters (water and snow/ice) in this study [43, 44].

8. Rainfall, especially intensive rain or heavy rain, is among the most significant inducing factors of HLDs [14]. Mean precipitation is defined as the annual accumulative rainfall values and the data can be obtained from the China Meteorological Science Data Sharing Network (http://data.cma.cn) [45].

9. Profile curvature is defined as the curvature in the downslope direction along a line formed by the intersection of an imaginary vertical plane with the ground surface [14], which is widely used in LSM.

10. *SPI* index has very important effects on the formulations of HLDs. The calculation method of *SPI* is showed in Eq (1).

$$SPI = A_s \cdot \tan\beta \qquad (1)$$

Where $A_s$ is the specific catchment area, and $\beta$ (radians) is the slope gradient [14, 46].

11. *TWI* index is defined as the function of both the slope and upstream contributing area per unit width orthogonal to the flow direction [14, 47]. The calculation method of *TWI* is showed in Eq (2).

$$TWI = \ln(A_s/\tan\beta) \qquad (2)$$

*TWI* is actually a quantitative description of the length of the runoff path, the area of the runoff, and so on. It is a quantification of the potential (theoretical) soil moisture content and potential capacity of runoff at various points in the basin [48].

## 2.2 HLDs inventory

In order to further define the disaster pregnant environment and occurring regulations of HLDs, and provide a database for subsequent calculations, the HLDs inventory was compiled by combining the field survey, visual interpretation of satellite images or aerial photographs and historical reports [1, 3, 49, 50]. 1543 disaster points and 1543 non-disaster points along 9 expressways, 15 national highways and 8 provincial highways in 15 provinces were investigated. Investigation contents included the stake numbers and values of impact factors of each disaster point and non-disaster point [1, 3]. An overview of the highway segments in the HLDs inventory is showed in Table 1, some representative disaster points are showed in Fig 2.

According to the findings in the investigation, basic occurring regulations of HLDs can be summarized as below

HLDs generally occur on slopes exceed 25˚, the time of occurrence is approximately 2 hours after the start of rainfall to 5 days after the end of rainfall. The mean precipitation in disaster concentration areas generally exceeds 900mm and the annual average rainstorm days exceed 6.

**Table 1. Highway segments in the HLDs inventory.**

| Highway segments | Quantities of disaster points and non-disaster points | Highway segments | Quantities of disaster points and non-disaster points |
|---|---|---|---|
| Shenda expressway (Liaoning) | 57/52 | G207 Baotou segment (Inner Mongolia) | 36/60 |
| Wukui expressway (Xinjiang) | 49/47 | G210 Yulin segment (Shaanxi) | 42/58 |
| Binbo expressway (Shandong) | 44/41 | G210 Dazhou segment (Sichuang) | 55/45 |
| Xihan expressway (Shaanxi) | 38/60 | G213 Wenchuan segment (Sichuan) | 62/61 |
| Yonglan expressway (Hunan) | 61/37 | G219 Pishan segment (Xinjiang) | 34/37 |
| Chengya expressway (Sichuang) | 37/39 | G310 Shangluo segment (Shaanxi) | 39/41 |
| Zhangwu expressway (Fujian) | 41/45 | G321 Mianyang segment (Sichuang) | 42/46 |
| Duzhi expressway (Guizhou) | 54/51 | G338 Hanzhong segment (Shaanxi) | 47/54 |
| Kaihe expressway (Yunnan) | 39/39 | Beijing S109 Mentouggou segment | 47/57 |
| G104 Sanming segment (Fujian) | 44/57 | Liaoning S214 Tieling segment | 57/60 |
| G106 Huanggang segment (Hubei) | 43/47 | Shandong S236 Yiyuan segment | 64/57 |
| G108 Taiyuan segment (Shanxi) | 38/34 | Shaanxi S206 Jingbian segment | 37/41 |
| G108 Hanzhong segment (Shaanxi) | 64/64 | Shaanxi S302 Yuyang segment | 50/42 |
| G110 Yinchuang segment (Ningxia) | 47/37 | Guizhou S312 Anshun segment | 61/48 |
| G201 Changbai segment (Jilin) | 36/42 | Jiangxi S102 Nanchang segment | 54/47 |
| G205 Nanping segment (Fujian) | 75/43 | Fujian S302 Nanping segment | 49/54 |

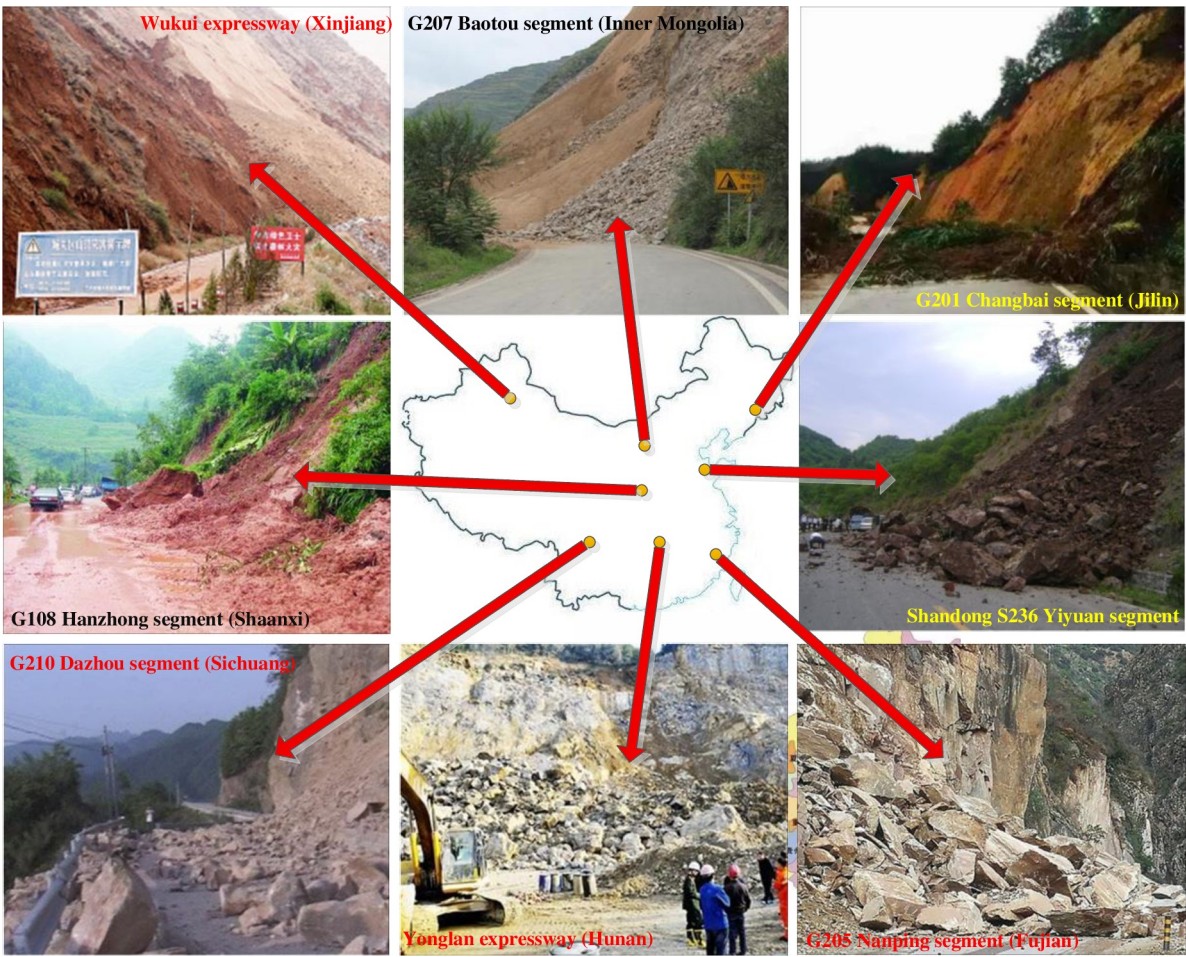

**Fig 2. Representative disaster points.**

The totally volume of the 1543 HLDs is about $8.3 \times 10^6$ m$^3$, differences in scales of HLDs are large, ranging from 12 m$^3$ to $9.6 \times 10^4$ m$^3$. The lithology that easily results in HLDs include silt, loess, clastic rock, mud rock, soft and flake metamorphic rock, shale, slate, soft stratum, argillization stratum and tectonically fractured stratum [44–46].

Earthquakes result in loosening of the mountains and provide massive loose deposits, so HLDs in the Wenchuan, Yushu and Ya'an earthquake areas are relatively more serious and the densities and scales of HLDs have significant positive correlations with the earthquake intensities [7, 16].

## 2.3 Data preparation

Eq (3) was implemented to normalize the values of the impact factors of HLDs.

$$x_i^* = \frac{x_i - x_{\min}}{x_{\max} - x_{\min}} \tag{3}$$

Where $x_i^*$ and $x_i$ indicate the normalized and original values of each impact factor, $x_{\max}$ and $x_{\min}$ indicate the maximum and minimum values of each impact factor. For quantitative factors, $x_i$, $x_{\max}$ and $x_{\min}$ were assigned with the values obtained directly from the HLDs inventory. For qualitative factors (lithology and land use), the classification assignment method was

implemented, i.e., lithology were classified into 8 types and the values were 1 for extremely hard rock, 2 for secondary hard rock, 3 for extremely soft rock, 4 for gravel soil, 5 for cohesive soil, 6 for sandy soil, 7 for silty soil and 8 for loess; there were 7 types of land use and the values were 1 for bareland, 2 for cropland, 3 for forest, 4 for grassland, 5 for residential land, 6 for wetland and 7 for waters.

The Spearman's rank correlation coefficient $r(X, Y)$ is a statistical factor that reflects the closeness of correlation between variables $X$ and $Y$ [51, 52], the calculation method is showed in Eq (4).

$$r(X, Y) = \frac{\text{cov}(X, Y)}{\sqrt{Var[X]}\sqrt{Var[Y]}}$$ (4)

Where $\text{cov}(X, Y)$ is the covariance of $X$ and $Y$, $Var[X]$ and $Var[Y]$ are the variances of $X$ and $Y$. The relationships between the degree of linear correlation and $r(X, Y)$ are summarized as follows [39]:

$$\begin{cases} |r(X, Y)| \leq 0.3 & \text{no linear correlation;} \\ 0.3 < |r(X, Y)| \leq 0.5 & \text{low linear correlation;} \\ 0.5 < |r(X, Y)| \leq 0.7 & \text{significant linear correlation;} \\ |r(X, Y)| \geq 0.7 & \text{highly linear correlation.} \end{cases}$$ (5)

The correlation coefficient matrix of the impact factors of HLDs was gained based on the analysis of the HLDs inventory, as showed in Table 2.

As showed in Table 2, there exists highly linear correlation and significant linear correlation among multiple couples of impact factors [53], and it is reasonable and feasible to extract the susceptibility evaluation indexes of HLDs according to PCA method.

## 2.4 Results of PCA method

PCA method is a traditional statistical analysis method and mainly used to deal with data with high dimensions and good correlations between variables, which can transform multiple factors into a few comprehensive factors. The principal components are defined as the unit

**Table 2. Correlation coefficient matrix.**

| | $n_1$ | $n_2$ | $n_3$ | $n_4$ | $n_5$ | $n_6$ | $n_7$ | $n_8$ | $n_9$ | $n_{10}$ | $n_{11}$ | $n_{12}$ |
|---|---|---|---|---|---|---|---|---|---|---|---|---|
| $n_1$ | 1 | | | | | | | | | | | |
| $n_2$ | 0.285 | 1 | | | | | | | | | | |
| $n_3$ | 0.463 | 0.105 | 1 | | | | | | | | | |
| $n_4$ | 0.384 | 0.094 | 0.824 | 1 | | | | | | | | |
| $n_5$ | 0.076 | -0.157 | 0.174 | 0.093 | 1 | | | | | | | |
| $n_6$ | -0.034 | -0.063 | 0.346 | 0.164 | 0.017 | 1 | | | | | | |
| $n_7$ | 0.029 | 0.106 | -0.174 | -0.128 | -0.141 | -0.026 | 1 | | | | | |
| $n_8$ | 0.104 | 0.095 | 0.183 | -0.082 | 0.124 | 0.252 | 0.016 | 1 | | | | |
| $n_9$ | 0.031 | -0.042 | 0.093 | 0.056 | 0.713 | -0.161 | 0.183 | 0.626 | 1 | | | |
| $n_{10}$ | -0.056 | -0.074 | 0.123 | 0.179 | -0.034 | 0.084 | 0.123 | 0.673 | 0.731 | 1 | | |
| $n_{11}$ | -0.057 | 0.062 | 0.256 | 0.037 | -0.026 | 0.582 | 0.031 | -0.162 | -0.026 | -0.064 | 1 | |
| $n_{12}$ | -0.418 | -0.136 | -0.625 | -0.683 | 0.073 | -0.203 | -0.027 | 0.284 | 0.083 | 0.194 | 0.037 | 1 |

$n_1$ represents slope, $n_2$ represents elevation, $n_3$ represents slope aspect, $n_4$ represents lithology, $n_5$ represents distance to faults, $n_6$ represents distance to rivers, $n_7$ represents NDVI, $n_8$ represents land use, $n_9$ represents mean precipitation, $n_{10}$ represents profile curvature, $n_{11}$ represents SPI and $n_{12}$ represents TWI.

orthogonal eigenvectors corresponding to the eigenvalues of the covariance matrix, from the view of mathematics, solving the principal components equivalents to solve the characteristic roots and eigenvectors according to the covariance matrix of the data source [54, 55]. which can be represented by the linear combination of the covariance matrix and original variables, as showed in Eq (6).

$$
\begin{cases}
Y_1 = \mu_{11}X_1 + \mu_{21}X_2 + \mu_{31}X_3 + \cdots + \mu_{p1}X_p \\
Y_2 = \mu_{12}X_1 + \mu_{22}X_2 + \mu_{32}X_3 + \cdots + \mu_{p2}X_p \\
\qquad \cdots\cdots\cdots \\
Y_p = \mu_{1p}X_1 + \mu_{2p}X_2 + \mu_{3p}X_3 + \cdots + \mu_{pp}X_p
\end{cases}
\tag{6}
$$

Where $Y_i$ is the principal component, $\mu_{ij}$ is the element of the covariance matrix, $X_j$ is the original variable. Usually, only principal components with large variances are selected to simplify the system structure. The concept of contribution rate is introduced in Eq (7).

$$
P_k = \lambda_k / \sum_{i=1}^{p} \lambda_i
\tag{7}
$$

Where $\lambda_i$ is the characteristic root of the covariance matrix, $P_k$ is the contribution rate of the $k$th characteristic root [55]. Eigenvectors, eigenvalues, contribution rates and accumulative contribution rate of the principal components ($F_1$, $F_2$, $F_3$ and $F_4$) corresponding to the normalized impact factor values of HLDs were calculated upon the principle of eigenvalues great than 1 and accumulative contribution rate great than 85%, as showed in Table 3.

As showed in Table 3, the contribution rates of $F_1$, $F_2$, $F_3$ and $F_4$ are 47.622%, 21.425%, 13.905% and 9.098% respectively and the accumulative contribution rate is 92.050%. Among them, $F_1$ mainly indicates the elevation, land use, mean precipitation and profile curvature factors; $F_2$ mainly indicates the slope, slope aspect and lithology factors; $F_3$ mainly indicates the distance to faults, distance to rivers and $SPI$ factors; $F_4$ mainly indicates the NDVI and $TWI$

**Table 3. Results of PCA method.**

| Principal components | | $F_1$ | $F_2$ | $F_3$ | $F_4$ |
|---|---|---|---|---|---|
| Eigenvectors | $n_1$ | 0.184 | -0.684 | 0.254 | 0.269 |
| | $n_2$ | 0.674 | 0.074 | -0.274 | 0.182 |
| | $n_3$ | 0.058 | 0.863 | -0.138 | -0.122 |
| | $n_4$ | -0.036 | 0.976 | 0.034 | 0.251 |
| | $n_5$ | 0.273 | -0.164 | 0.946 | 0.163 |
| | $n_6$ | -0.178 | 0.269 | 0.835 | -0.084 |
| | $n_7$ | -0.022 | 0.202 | -0.205 | 0.832 |
| | $n_8$ | 0.942 | -0.096 | -0.164 | -0.153 |
| | $n_9$ | 0.737 | 0.286 | 0.046 | -0.064 |
| | $n_{10}$ | 0.845 | -0.143 | 0.186 | -0.124 |
| | $n_{11}$ | 0.152 | 0.276 | -0.795 | 0.096 |
| | $n_{12}$ | -0.032 | 0.134 | 0.375 | 0.946 |
| Eigenvalues | | 6.726 | 3.026 | 1.964 | 1.285 |
| Contribution rates/% | | 47.622 | 21.425 | 13.905 | 9.098 |
| Accumulative contribution rate/% | | 47.622 | 69.047 | 82.952 | 92.050 |

factors. The calculation methods of $F_1$, $F_2$, $F_3$ and $F_4$ are showed in Eqs (8)–(11).

$$F_1 = 0.184n_1 + 0.674n_2 + 0.058n_3 - 0.036n_4 + 0.273n_5 - 0.178n_6$$
$$- 0.022n_7 + 0.942n_8 + 0.737n_9 + 0.845n_{10} + 0.152n_{11} - 0.032n_{12}$$
$$(8)$$

$$F_2 = -0.684n_1 + 0.074n_2 + 0.863n_3 + 0.976n_4 - 0.164n_5 + 0.269n_6$$
$$+ 0.202n_7 - 0.096n_8 + 0.286n_9 - 0.143n_{10} + 0.276n_{11} + 0.134n_{12}$$
$$(9)$$

$$F_3 = 0.254n_1 - 0.274n_2 - 0.138n_3 + 0.034n_4 + 0.946n_5 + 0.835n_6$$
$$- 0.205n_7 - 0.164n_8 + 0.046n_9 + 0.186n_{10} - 0.795n_{11} + 0.375n_{12}$$
$$(10)$$

$$F_4 = 0.269n_1 + 0.182n_2 - 0.122n_3 + 0.251n_4 + 0.163n_5 - 0.084n_6$$
$$+ 0.832n_7 - 0.153n_8 - 0.064n_9 - 0.124n_{10} + 0.096n_{11} + 0.946n_{12}$$
$$(11)$$

## 3 Susceptibility evaluation methods

### 3.1 Evaluation models

SVM model was first introduced by Boser, Guyon and Vapnik in 1992. By employing a learning algorithm relying on statistical learning theory and optimization theory, SVM enables the computer to learn how to implement classification and regression tasks, increase prediction accuracy, and also avoid over fitting drawbacks. SVM is popular for its better empirical performance compared to sophisticated neural network functions, easy training process, avoiding local minima, relatively suitable mathematics for high dimensional data and finding the best trade-off between complexity (over generalization) and error (over fitting) [56]. The Gauss Radial Basis Function was introduced to SVM model for susceptibility evaluation of HLDs in this study, which selected 70% disaster points and 70% non-disaster points in the HLDs inventory as the network training samples, the remaining 30% disaster points and 30% non-disaster points as the verification samples, and the values of the principal components as the network input and the occurring probabilities of HLDs as the output (with values from 0 to 1, 0 indicates the disaster will not occur and 1 indicates the disaster will occur inevitably). In order to improve the evaluation efficiency and calculation accuracy, PSO model and GA model were implemented to search the optimum values of the penalty parameter $C$ and nuclear parameter $\sigma$ respectively [57].

**3.1.1 PSO-SVM model.** The processes of susceptibility evaluation of HLDs through the PSO-SVM model are showed in Fig 3 [57].

Detailed modeling methods are showed as below [58]:

1. Set initial parameters of the PSO model to generate random initial particles and initial speeds of the particles; set population size to 20, evolving algebra $k$ to 100, learning factors $c_1$ and $c_2$ to 2.05 and 2.35, inertia weight $\omega$ to 0.5, optimization scope of the penalty parameter $C$ to (0, 100] and nuclear parameter to (0, 1000].

2. The processes of parameter optimization were the training processes of the SVM network. During optimization, each solution of the optimization problem was considered as a particle in the solution space. Each $C$ and $\sigma$ corresponding to an SVM network and the particles were measured and evaluated upon fitness.

3. Each particle was considered as one unit, the current position of each particle $f_i$, the best position of each particle $q_i$ and the best position of the whole population $q_g$ were calculated by the fitness function; the speeds and positions of the particles were updated by comparing

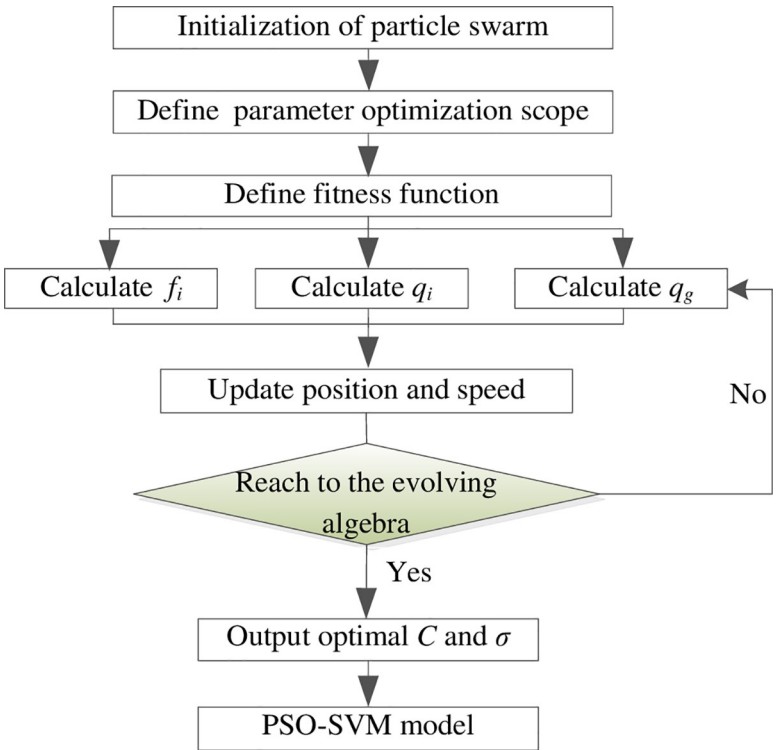

**Fig 3. PSO-SVM modeling processes.**

$f_i$, $q_i$ and $q_g$. If $f_i<q_i$, $q_i$ substituted $f_i$ as the best position of a particle; if $q_i<q_g$, $q_g$ substituted $q_i$ as the best position of the whole population. See Eq (12) for updating speeds and positions of the particles.

$$\begin{cases} v_i^{k+1} = \omega v_i^k + c_1 r_1 (q_i - x_i^k) + c_2 r_2 (q_g - x_i^k) \\ x_i^{k+1} = x_i^k + v_i^{k+1} \end{cases} \tag{12}$$

Where $i$ indicates the serial number of the particles, $r_1$ and $r_2$ indicate random numbers from 0 to 1, $v_i^k$ and $v_i^{k+1}$ indicate the flying speeds of the $i$th particle under $k$ and $k+1$ generations, $x_i^k$ and $x_i^{k+1}$ indicate the positions of the $i$th particle under $k$ and $k+1$ generations respectively.

4. The operation ended when the evolving algebra reached 100, the optimal fitness of the particle tended to be stable after the 22nd generation and the difference between the particle fitness and optimal fitness for the 4th generation was the minimum to get $C_{\text{optimal}} = 2.301$ and $\sigma_{\text{optimal}} = 6.284$. See Fig 4 for the particle fitness and optimal fitness.

5. $C = 2.301$ and $\sigma = 6.284$ were considered as the optimal parameter combination to build the PSO-SVM model, the verification samples were evaluated and the occurring probabilities were output.

**3.1.2 GA-SVM model.** The processes of susceptibility evaluation of HLDs through the GA-SVM model are showed in Fig 5 [59].

Detailed modeling methods are showed as below [60, 61]:

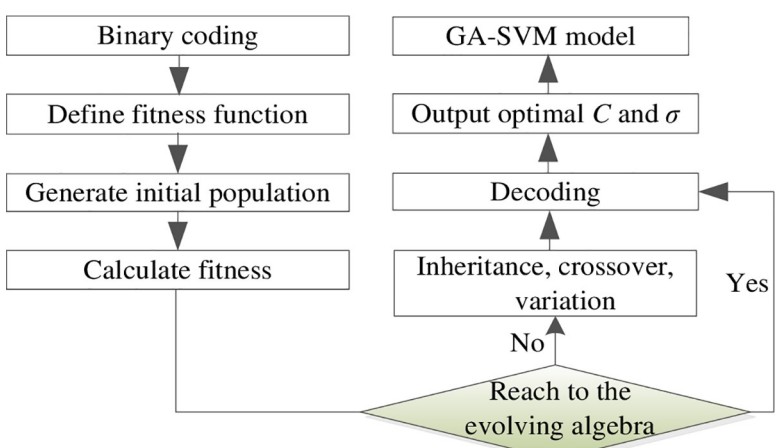

**Fig 4. Fitness curve of the PSO model.**

**Fig 5. GA-SVM modeling processes.**

1. Set initial parameters of the GA model to generate random initial population, set population size to 20, evolving algebra $k$ to 100, crossover probability to 0.9, variation probability to 0.1, optimization scope of the penalty parameter $C$ and nuclear parameter $\sigma$ to (0, 100]. As each piece of the chromosome consists of 10 genes, the total number of optional genes is 1024 and the optimization step length is 100/1024. For example, "0100000010" refers to the 130th chromosome and its value is 13000/1024.

2. Similarly, the processes of parameter optimization were the training processes of the SVM network and the mean square error (MSE) of the verification samples was defined as the fitness of the GA network. The fitness of each generation and the optimal fitness were calculated, inheritance, crossover and variation algorithms were implemented to search the new population in order to improve the calculation efficiency. The operation ended when it inherits to the 100th generation.

3. The optimal fitness tended to be stable after the 8th generation and there was the minimum difference between the particle fitness and optimal fitness for the 63rd generation to get $C_{\text{optimal}}$ = 25.391 and $\sigma_{\text{optimal}}$ = 1.465. See Fig 6 for the particle fitness and optimal fitness.

4. $C$ = 25.391 and $\sigma$ = 1.465 were considered as the optimal parameter combination to build the GA-SVM model, the verification samples were evaluated and the occurring probabilities were output.

## 3.2 Results of AUC method

AUC method was utilized to verify the evaluation results of the PSO-SVM model and GA-SVM model, which referred to normalize the occurring probabilities of the verification samples to 100 grades and sorted in descending order, the accumulative frequencies of disasters occurring within each grade were calculated and a curve was generated. The larger areas under the curve (AUC value) indicate more accurate evaluation results, when the AUC value is 1, the evaluation results are completely correct [62]. According to the verification results, the AUC value of the PSO-SVM model is 0.907, the success rate of the evaluation results is 0.846 for the top 10 grades and 0.891 for the top 20 grades. The AUC value of the GA-SVM model is 0.894, the success rate of the evaluation results is 0.725 for the top 10 grades and 0.839 for the top 20 grades. As a result, the evaluation results of the PSO-SVM model are better than those of the GA-SVM model, as showed in Fig 7.

## 4 Susceptibility mapping and zoning of HLDs

### 4.1 Susceptibility mapping of HLDs

In this study, the resolution of the impact factors of HLDs was set to 100 m×100 m in order to run the models. The distribution maps of the impact factors were overlapped upon Eqs (8)–(11) based on GIS to get the distribution of each evaluation index, where, the values of $F_1$ were -0.549–4.876, $F_2$ were -0.633–2.581, $F_3$ were -0.942–1.937 and $F_4$ were -0.672–2.762, as showed in Figs 8–11.

According to the PSO-SVM model and Figs 8–11, secondary development for GIS platform was conducted and the occurring probability distribution map of HLDs in China was plotted, as showed in Fig 12. The minimum and maximum occurring probabilities of HLDs in China are 0.092 and 0.837 respectively. The comprehensive distribution features indicate that higher susceptible levels in southeast China and lower susceptible levels in northwest China. Areas

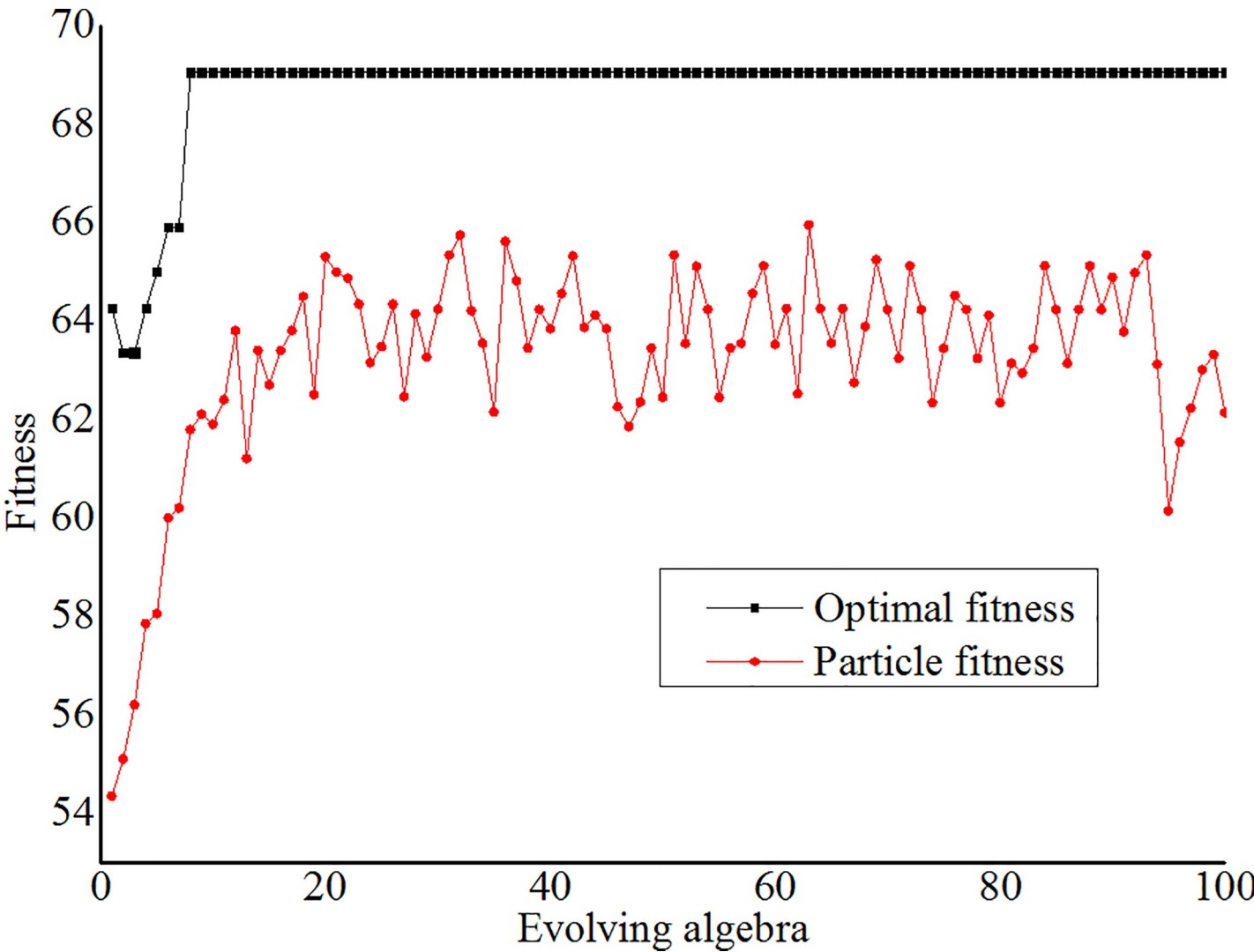

**Fig 6. Fitness curve of the GA model.**

with low occurring probabilities include east Northeast China Plain, Inner Mongolian Plateau, Sinkiang Basin and north Qinghai- Tibet Plateau. Areas with high occurring probabilities include eastern mountain areas of Zhejiang and Fujian, Taiwan Mountain, Qinling-Daba Mountain, Kunlun Mountain, Tianshan Mountain, Hengduan Mountain and east Qinghai-Tibet Plateau.

### 4.2 Susceptibility zoning of HLDs

Considering the occurring probabilities of HLDs as the dominant index as well as the zoning boundaries of other natural disasters in China, four susceptible levels and 14 dangerous areas of HLDs were regionalized. The occurring probability classification standards are as follows: extreme dangerous: 0.651–0.837; severe dangerous: 0.464–0.651; moderate dangerous: 0.278–0.464; micro dangerous: 0.092–0.278. The susceptibility zoning map of HLDs in China was

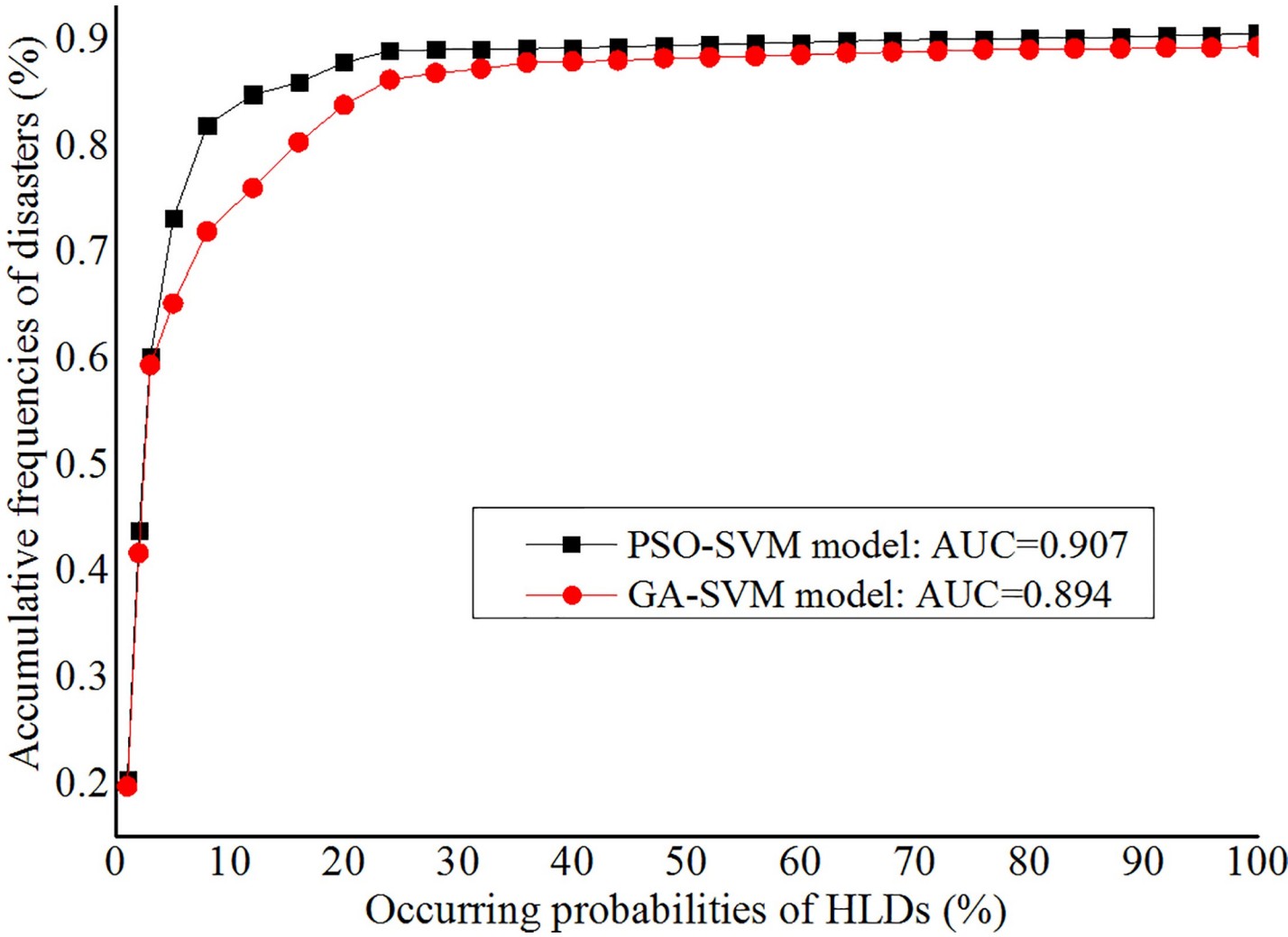

**Fig 7. Verification results of the PSO-SVM model and GA-SVM model.**

plotted based on GIS and the corresponding susceptibility zoning scheme was formulated, as showed in Fig 13 and Table 4.

As showed in Fig 13 and Table 4, the extreme dangerous areas include Sichuan, Yunnan and Guizhou Mountain- Hengduan Mountain-Qinling-Daba Mountain, East Zhejiang-Wuyi Mountain-Nanling Mountain-Taiwan Mountain and Tianshan-Kunlun Mountain, which is consistent with the actual distribution conditions of HLDs indicated upon decades of highway construction experience. Among the 1543 landslides in the HLDs inventory, there are 806 located in the extreme dangerous areas and 421 located in the severe dangerous areas, accounting for 52.23% and 27.28% respectively, while the extreme dangerous areas and severe dangerous areas account for only 19.74% and 36.53% of the total areas of China. There are 182 and 134 HLDs in the moderate dangerous areas and micro dangerous areas, accounting for 11.81% and 8.68% respectively, while the moderate dangerous areas and micro dangerous areas account for 19.49% and 24.24% of the total areas of China. As a result, the susceptibility zoning scheme of HLDs in China is scientific and reasonable.

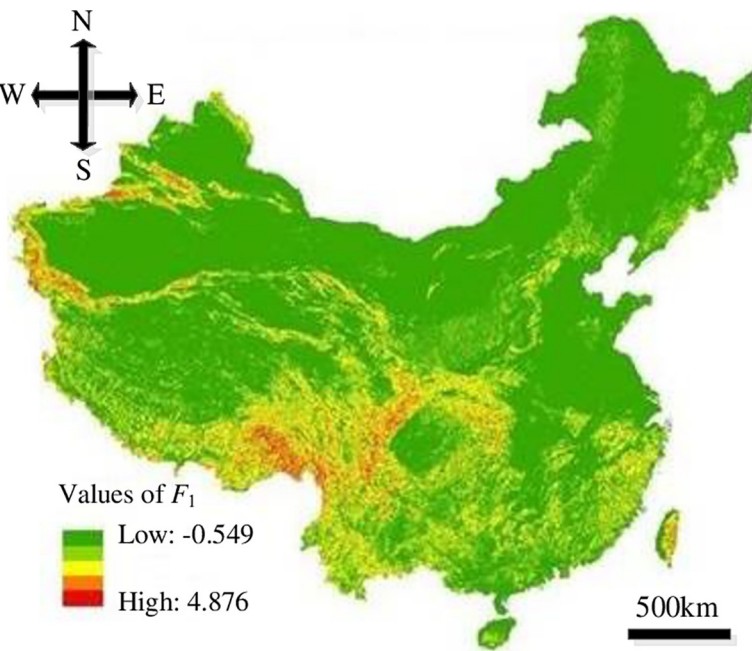

**Fig 8. Distribution of $F_1$.**

## 5 Conclusions

1. Impact factors of HLDs included slope, elevation, slope aspect, lithology, distance to faults, distance to rivers, NDVI, land use, mean precipitation, profile curvature, *SPI* and *TWI*. The HLDs inventory containing 1543 disaster points and 1543 non-disaster points along 9

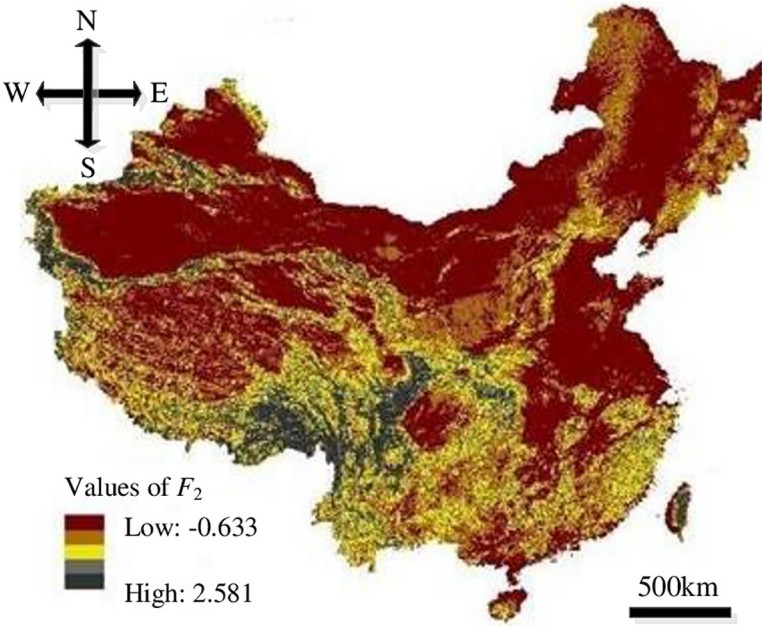

**Fig 9. Distribution of $F_2$.**

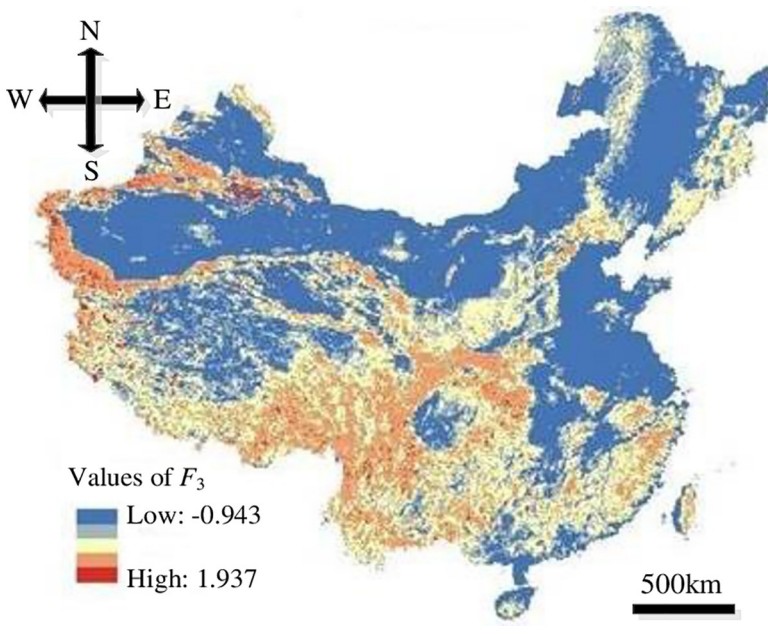

**Fig 10. Distribution of $F_3$.**

expressways, 15 national highways and 8 provincial highways in 15 provinces was compiled. PCA method was implemented to extract the susceptibility evaluation indexes and four principal components were obtained, whose accumulative contribution rate was 92.050%. The PSO-SVM model and GA-SVM model were used to susceptibility evaluation of HLDs in China respectively, the evaluation results of the PSO-SVM model were better than those

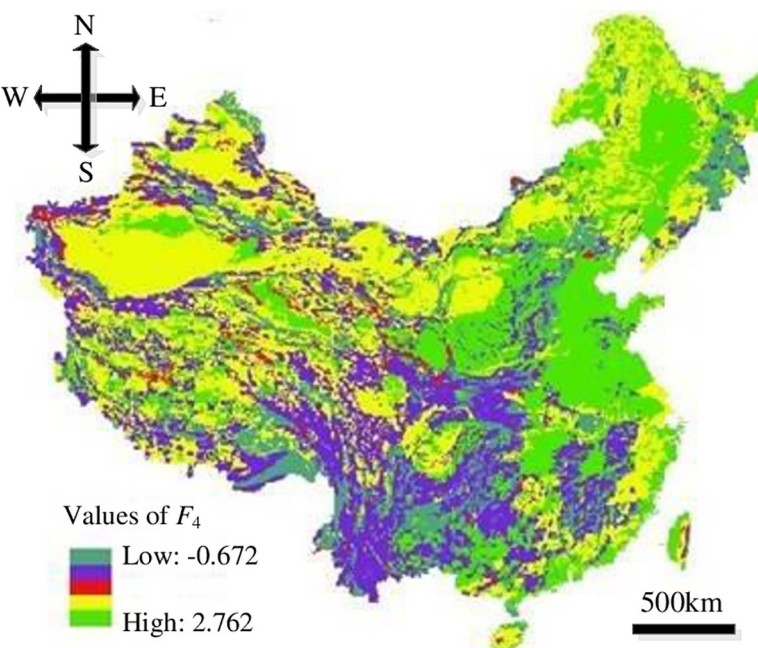

**Fig 11. Distribution of $F_4$.**

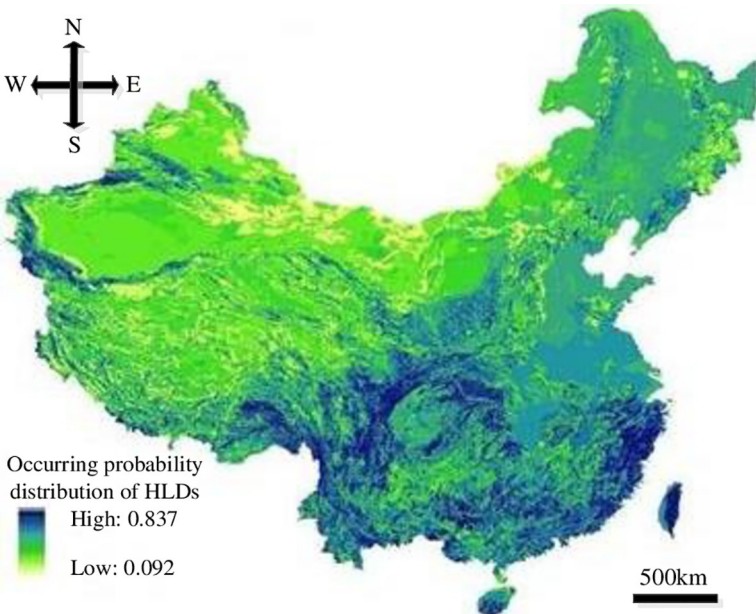

**Fig 12. Occurring probability distribution map of HLDs in China.**

of the GA-SVM model. Micro dangerous areas, moderate dangerous areas, severe dangerous areas and extreme dangerous areas accounted for 24.24%, 19.49%, 36.53% and 19.74% of the total areas of China, among the 1543 disaster points in the HLDs inventory, there were 134, 182, 421 and 806 located in the above areas respectively.

2. This study can be improved from several aspects as below: (1) The evaluation results of the PSO-SVM model are better than those of the GA-SVM model, but the AUC value was only

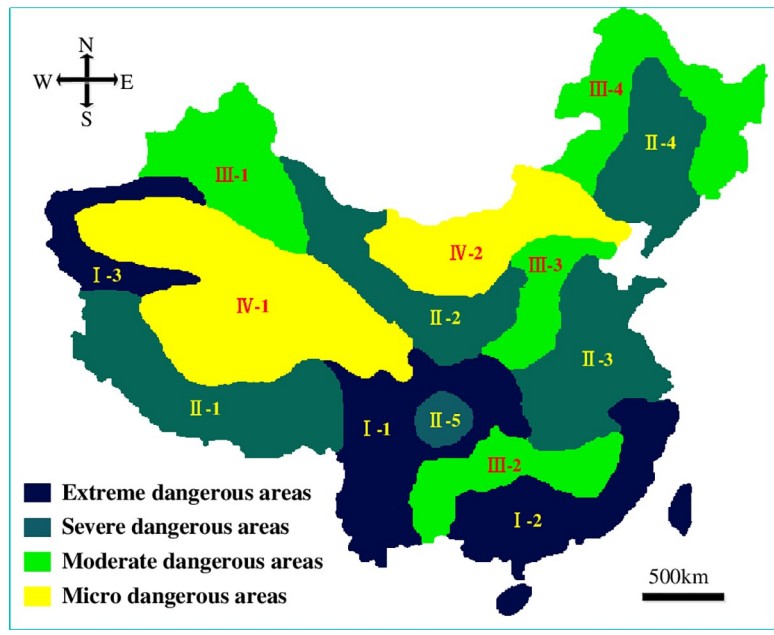

**Fig 13. Susceptibility zoning map of HLDs in China.**

**Table 4. Susceptibility zoning scheme of HLDs in China.**

| Susceptible levels | Susceptibility zoning scheme of HLDs |
|---|---|
| I: Extreme dangerous | I-1: Sichuan, Yunnan and Guizhou Mountain-Hengduan Mountain-Qinling-Daba Mountain |
| | I-2: East Zhejiang-Wuyi Mountain-Nanling Mountain-Taiwan Mountain |
| | I-3: Tianshan Mountain-Kunlun Mountain |
| II: Severe dangerous | II-1 South Qinghai-Tibet Plateau |
| | II-2 West Inner Mongolian Plateau-Hetao Region-Loess Plateau |
| | II-3 North China Plain-Jianghuai Plain-Middle and Lower Reaches of the Yangtze River Plain |
| | II-4 Northeast Plain-Changbai Mountain |
| | II-5 Sichuan Basin |
| III: Moderate dangerous | III-1 Tarim Basin-Altai Mountain |
| | III-2 East Yunnan-North Nanling |
| | III-3 Taihang Mountain |
| | III-4 Great Khingan-East Northeast Plain |
| IV: Micro dangerous | IV-1 Qaidam Basin-North Inner Mongolia Plateau |
| | IV-2 Middle and east Inner Mongolia Plateau |

0.907 and the evaluation accuracy could be further improved. In addition, other evaluation methods such as the LR, ANN and information value method were not implemented and their evaluation accuracies were not verified; (2) The occurring probabilities of HLDs were considered as the dominant index of susceptibility zoning and the zoning boundaries were determined upon isometric principle, which decreased the accuracies of the susceptibility zoning results to some extent. Studies that determines the susceptibility zoning boundaries based on the cluster analysis has not been developed.

## Supporting information

**S1 File.**
(XLSX)

**S2 File.**
(XLSX)

**S3 File.**
(XLSX)

**S4 File.**
(XLSX)

**S5 File.**
(XLS)

## Author Contributions

**Conceptualization:** Fa Che.

**Data curation:** Chao Yin, Haoran Li.

**Formal analysis:** Ying Li, Zhinan Hu, Dong Liu.

**Funding acquisition:** Chao Yin, Fa Che.

**Investigation:** Zhinan Hu.

**Methodology:** Chao Yin, Haoran Li, Ying Li.

**Project administration:** Chao Yin, Fa Che, Dong Liu.

**Resources:** Fa Che, Ying Li.

**Software:** Chao Yin, Zhinan Hu.

**Validation:** Chao Yin, Dong Liu.

**Visualization:** Fa Che, Ying Li, Zhinan Hu.

**Writing – original draft:** Chao Yin, Haoran Li.

**Writing – review & editing:** Chao Yin, Haoran Li, Fa Che.

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
