## [Decision Letter · Decision Letter 0]

16 Apr 2020

PONE-D-20-03927

Hazard assessment and regionalization of highway landslide disasters based on the optimized support vector machine

PLOS ONE

Dear Dr. Yin,

Thank you for submitting your manuscript to PLOS ONE. After careful consideration, we feel that it has merit but does not fully meet PLOS ONE’s publication criteria as it currently stands. Therefore, we invite you to submit a revised version of the manuscript that addresses the points raised during the review process.

I recommend that all notes posted by the reviewers are taken care of.

We would appreciate receiving your revised manuscript by May 31 2020 11:59PM. To enhance the reproducibility of your results, we recommend that if applicable you deposit your laboratory protocols in protocols.io, where a protocol can be assigned its own identifier (DOI) such that it can be cited independently in the future. For instructions see: http://journals.plos.org/plosone/s/submission-guidelines#loc-laboratory-protocols

We look forward to receiving your revised manuscript.

Kind regards,

Claudionor Ribeiro da Silva

Academic Editor

PLOS ONE

Journal Requirements:

1. Thank you for including your competing interests statement; "The authors have declared that no competing interests exist."

We note that one or more of the authors are employed by a commercial company:  “Urban Rail Construction Corporation, Zhongtian Construction Group Co., LTD”

Reviewers' comments:

Reviewer's Responses to Questions

**Comments to the Author**

1. Is the manuscript technically sound, and do the data support the conclusions?

Reviewer #1: Partly

Reviewer #2: Yes

Reviewer #3: No

2. Has the statistical analysis been performed appropriately and rigorously? 

Reviewer #1: N/A

Reviewer #2: Yes

Reviewer #3: No

3. Have the authors made all data underlying the findings in their manuscript fully available?

Reviewer #1: Yes

Reviewer #2: Yes

Reviewer #3: No

4. Is the manuscript presented in an intelligible fashion and written in standard English?

Reviewer #1: Yes

Reviewer #2: Yes

Reviewer #3: Yes

5. Review Comments to the Author

Reviewer #1: Totally the present article is well-established and the subject is interesting, but some major revision should be considered.

- More suitable title should be selected for the article. Title should decrease to 10-12 words.

- The abstract should state briefly the purpose of the research, the principal results and major conclusions. An abstract is often presented separately from the article, so it must be able to stand alone.

- It is suggested to present the structure of the article at the end of the introduction.

- The necessity and innovation of the article should be presented to the introduction.

- Literature review is not enough. There some articles, which must be added to literature review:

Li & Leao. Application of Nor Sand Constitutive Model in a Highway Fill Embankment Slope Stability Failure Study;

Emeka et al. Deformation behaviour of erodible soil stabilized with cement and quarry dust.

- A flowchart should be added to the article to show the research methodology.

- The quality of figures 7 to 11 is weak. The original source of the figures should be used into the manuscript.

- A map should be presented for the study area. It is suggested to show the general location and then in 2 or 3 step show the exact location.

- The major defect of this study is the debate or Argument is not clear stated in the introduction session. Hence, the contribution is weak in this manuscript. I would suggest the author to enhance your theoretical discussion and arrives your debate or argument.

- It is suggested to compare the results of the present research with some similar studies which is done before.

- I would suggest you to discuss regarding the supportvector machine method (you can use researches entitled “Application of Support Vector Machine and Gene Expression Programming on Tropospheric ozone Prognosticating for Tehran Metropolitan” and “A Modern Method to Improve of Detecting and Categorizing Mechanism for Micro Seismic Events Data Using Boost Learning System”).

- Much more explanations and interpretations must be added for the Results, which are not enough.

- Please make sure your conclusions' section underscore the scientific value added of your paper, and/or the applicability of your findings/results, as indicated previously. Please revise your conclusion part into more details. Basically, you should enhance your contributions, limitations, underscore the scientific value added of your paper, and/or the applicability of your findings/results and future study in this session.

- DOI of the references should be added (you can use “" ext-link-type="uri" xlink:type="simple">https://crossref.org/").

- “Notation” should be added to the article.

Reviewer #2: The article is very well written and the presented work is very interested. The particle swarm optimization optimized support vector machine (PSO-SVM) and genetic algorithm optimized support vector machine (GA-SVM) were used to hazard assessment.

Reviewer #3: This study aim to define the hazard spatial differentiation conditions of HLDs from the macro perspective using the principal component analysis (PCA) to extract the hazard assessment indexes known as particle swarm optimization optimized support vector machine (PSO-SVM) and genetic algorithm optimized support vector machine (GA-SVM) over the mainland of China.

In my opinion this work is not enough to publish for the following reason:

1- The number of observation data is not sufficient for such a big study area. there should be some justification for the observed landslides and simulated susceptible areas.

2- Observed landslide should be ranked based on the magnitude in terms of area, damages or the volume of mass movement to be use for assessment accuracy.

3- Base on my knowledge Topo map at scale 1:250000 is too coarse to derive slope or other topographic indices.

this scale may useful for site planning like suitable area for ground water recharge, but not useful to delineate risky area for highway construction which is considered as line in spatial date modeling.

4- There is no any accuracy assessment in your study.

5- Figure-11 (Hazard rationalization map of HLDs in China) sounds like hand drowning instead of GIS-based standard map classification.

6- All maps need to have some map annotations including Scale bar, North arrow, Coordinate system.

6. PLOS authors have the option to publish the peer review history of their article (what does this mean?). If published, this will include your full peer review and any attached files.

Reviewer #1: No

Reviewer #2: Yes: Afaq Ahmad

Reviewer #3: Yes: Dr Abolghasem Akbari

---

## [Author Response · Author response to Decision Letter 0]

26 May 2020

Response to Reviewer #1: 

 (1) More suitable title should be selected for the article. Title should decrease to 10-12 words.

 Response: The title of this manuscript was modified to “Susceptibility mapping and zoning of highway landslide disasters in China”, this title is more suitable for the contents of the manuscript.

 (2) The abstract should state briefly the purpose of the research, the principal results and major conclusions. An abstract is often presented separately from the article, so it must be able to stand alone.

 Response: The abstract of this manuscript was rewrote as follows: “Prominent regional differentiations of highway landslide disasters (HLDs) bring great difficulties in highway planning, designing and disaster mitigation, therefore, a comprehensive understanding of HLDs from the spatial perspective is a basis for reducing damages. Statistical prediction methods and machine learning methods have some defects in landslide susceptibility mapping (LSM), meanwhile, hybrid methods have been developed by combining the statistical prediction methods with machine learning methods in recent years, and some of them were reported to perform better than conventional methods. In view of this, the principal component analysis (PCA) method was used to extract the susceptibility evaluation indexes of HLDs; the particle swarm optimization-support vector machine (PSO-SVM) model and genetic algorithm-support vector machine (GA-SVM) model were implemented to the susceptibility mapping and zoning of HLDs in China. The research results show that the accumulative contribution rate of the four principal components is 92.050%; evaluation results of the PSO-SVM model are better than those of the GA-SVM model; micro dangerous areas, moderate dangerous areas, severe dangerous areas and extreme dangerous areas account for 24.24%, 19.49%, 36.53% and 19.74% of the total areas of China; among the 1543 disaster points in the HLDs inventory, there are 134, 182, 421 and 806 located in the above areas respectively.”

 (3) It is suggested to present the structure of the article at the end of the introduction.

 Response: The structure of this manuscript was presented at the end of the Introduction as follows: “The contents of this study include: (1) select the impact factors of HLDs; (2) compile the HLDs inventory; (3) extract the susceptibility evaluation indexes of HLDs by PCA method; (4) determine the better evaluation model by AUC method; (6) susceptibility mapping of HLDs; and (7) propose the susceptibility zoning scheme of HLDs in China”.

 (4) The necessity and innovation of the article should be presented to the introduction.

 Response: The necessity and innovation of this manuscript was presented as follows: “There are still several defects of current researches on LSM: (1) Current researches generally focus on the view of physical geography, however, this unprofessional mapping cannot reflect on the mutual feedback mechanism between the occurrences of HLDs and their disaster pregnant environment, only provide indirect references for highway planning, designing and disaster mitigation; (2) SVM is one of the main modeling methods implemented to LSM, the critical factors affect its calculation efficiency are the optimization speeds of the penalty parameter C and nuclear parameter σ, when the optimization scope is large, SVM often tends to consider the partial optimum as overall optimum, resulting in early maturity. Hybrid methods have been developed by combining the statistical prediction methods with machine learning methods in recent years, some of them were reported to perform better than conventional methods. In view of this, the principal component analysis (PCA) method was used to extract the susceptibility evaluation indexes of HLDs; the particle swarm optimization-support vector machine (PSO-SVM) model and genetic algorithm-support vector machine (GA-SVM) model were implemented to the susceptibility mapping and zoning of HLDs in China, and the better evaluation model was determined through the areas under curve (AUC) method.”

 (5) Literature review is not enough. There some articles, which must be added to literature review:

 Response: The literature review was rewrote and many new literatures were added including the articles proposed by the review, as follows: “Researches on landslide susceptibility mapping (LSM) in China mainly focused on the Wenchuan, Yushu and Ya’an earthquake areas, the Three Gorges Reservoir areas, the areas affected by typhoons and loess areas; researches abroad China mainly focused on the Medellin areas (Columbia), Kyushu areas (Japan) and some areas in Italy [13-15]. The modeling methods implemented to LSM mainly included the statistical prediction models, i.e., Logistic regression method (LR), decision tree method, analytical hierarchy process (AHP), deterministic coefficient method and multivariate adaptive regression spline model (MARSplines), and the machine learning models, i.e., artificial neural network (ANN), support vector machine (SVM), neuro-fuzzy technique, decision tree model and Bayesian network (BN), some scholars also conducted comparison researches on multiple modeling methods [11,16-20]. Representative studies included: Wang et al. [21] used the LR, bivariate statistical analysis (BS) and MARSplines to create landslide susceptibility maps by comparing the past landslide distribution and conditioning factor thematic maps; Alireza et al. [22] proposed a novel hybrid model based on the step-wise weight evaluation ratio analysis (SWARA) method and adaptive neuro-fuzzy inference system (ANFIS) to evaluate landslide susceptible areas using geographical information system (GIS); Zhang et al. [23] used the information value model and LR to build the susceptibility evaluation systems based on the data of 655 landslides in the history of Wanzhou district (Chongqing); Sezer et al. [24] conducted landslide susceptibility evaluation by applying the methods of M-AHP and Mamdani type FIS by using the expert-based LSM module; Chen et al. [25] built a landslide susceptibility model using three well-known machine learning models namely the maximum entropy (MaxEnt), SVM and ANN, and accompanied by their ensembles (i.e., ANN-SVM, ANN-MaxEnt, ANN-MaxEnt-SVM and SVM-MaxEnt) in Wanyuan (China); Zhu et al. [26] developed and compared two presence-only methods including the one-class SVM and kernel density estimation (KDE), and two presence-absence methods including the ANN and two-class SVM to evaluate their respective performance in mapping landslide susceptibility; Chen et al. [11] assessed and compared four advanced machine learning techniques, namely the BN, radical basis function classifier (RBF), logistic model tree (LMT) and random forest (RF) models, for landslide susceptibility modeling in Chongren, China; Yang et al. [27] proposed a new LSM method based on the GeoDetector and spatial logistic regression model (SLR), of which, the GeoDetector was used to select condition factors based on the spatial distribution of landslides, SLR model was used to make full use of the structural and attribute information of spatial objects simultaneously in LSM.”

 (6) A flowchart should be added to the article to show the research methodology.

 Response: The following flowchart was added to the manuscript. 

 Figure 1 Flowchart of this study

 (7) The quality of figures 7 to 11 is weak. The original source of the figures should be used into the manuscript.

 Response: These figures were modified as follows:

 Figure 8 Distribution of F1

 Figure 9 Distribution of F2

 Figure 10 Distribution of F3

 Figure 11 Distribution of F4

 Figure 12 Occurring probability distribution map of HLDs in China

 Figure 13 Susceptibility zoning map of HLDs in China

 (8) A map should be presented for the study area. It is suggested to show the general location and then in 2 or 3 step show the exact location.

 Response: A landslide inventory containing 1543 disaster points and 1543 non-disaster points along 9 expressways, 15 national highways and 8 provincial highways in 15 provinces was compiled in this study. It was difficult to present all the locations of these points in a single figure, so I added Table 1 and Figure 2 in the manuscript as follows, where Table 1 showed an overview of the highway segments in the HLDs inventory, Figure 2 showed some representative disaster points in the HLDs inventory.

Table 1 Highway segments in the HLDs inventory

Highway segments Quantities of disaster points and non-disaster points Highway segments Quantities of disaster points and non-disaster points

Shenda expressway (Liaoning) 57/52 G207 Baotou segment (Inner Mongolia) 36/60

Wukui expressway (Xinjiang) 49/47 G210 Yulin segment (Shaanxi) 42/58

Binbo expressway (Shandong) 44/41 G210 Dazhou segment (Sichuang) 55/45

Xihan expressway (Shaanxi) 38/60 G213 Wenchuan segment (Sichuan) 62/61

Yonglan expressway (Hunan) 61/37 G219 Pishan segment (Xinjiang) 34/37

Chengya expressway (Sichuang) 37/39 G310 Shangluo segment (Shaanxi) 39/41

Zhangwu expressway (Fujian) 41/45 G321 Mianyang segment (Sichuang) 42/46

Duzhi expressway (Guizhou) 54/51 G338 Hanzhong segment (Shaanxi) 47/54

Kaihe expressway (Yunnan) 39/39 Beijing S109 Mentougou segment 47/57

G104 Sanming segment (Fujian) 44/57 Liaoning S214 Tieling segment 57/60

G106 Huanggang segment (Hubei) 43/47 Shandong S236 Yiyuan segment 64/57

G108 Taiyuan segment (Shanxi) 38/34 Shaanxi S206 Jingbian segment 37/41

G108 Hanzhong segment (Shaanxi) 64/64 Shaanxi S302 Yuyang segment 50/42

G110 Yinchuang segment (Ningxia) 47/37 Guizhou S312 Anshun segment 61/48

G201 Changbai segment (Jilin) 36/42 Jiangxi S102 Nanchang segment 54/47

G205 Nanping segment (Fujian) 75/43 Fujian S302 Nanping segment 49/54

 Figure 2 Representative disaster points

 (9) The major defect of this study is the debate or Argument is not clear stated in the introduction session. Hence, the contribution is weak in this manuscript. I would suggest the author to enhance your theoretical discussion and arrives your debate or argument.

 Response: The debate and argument were added in the Introduction as follows: “There are still several defects of current researches on LSM: (1) Current researches generally focus on the view of physical geography, however, this unprofessional mapping cannot reflect on the mutual feedback mechanism between the occurrences of HLDs and their disaster pregnant environment, only provide indirect references for highway planning, designing and disaster mitigation; (2) SVM is one of the main modeling methods implemented to LSM, the critical factors affect its calculation efficiency are the optimization speeds of the penalty parameter C and nuclear parameter σ, when the optimization scope is large, SVM often tends to consider the partial optimum as overall optimum, resulting in early maturity. Hybrid methods have been developed by combining the statistical prediction methods with machine learning methods in recent years, some of them were reported to perform better than conventional methods.”

 (10) I would suggest you to discuss regarding the support vector machine method.

 Response: Some introduction of SVM model was added in the manuscript as follows: “SVM model was first introduced by Boser, Guyon and Vapnik in 1992. By employing a learning algorithm relying on statistical learning theory and optimization theory, SVM enables the computer to learn how to implement classification and regression tasks, increase prediction accuracy, and also avoid over fitting drawbacks. SVM is popular for its better empirical performance compared to sophisticated neural network functions, easy training process, avoiding local minima, relatively suitable mathematics for high dimensional data, and finding the best trade-off between complexity (over generalization) and error (over fitting).”

 (11) Please make sure your conclusions section underscore the scientific value added of your paper, and/or the applicability of your findings/results, as indicated previously. Please revise your conclusion part into more details. Basically, you should enhance your contributions, limitations, underscore the scientific value added of your paper, and/or the applicability of your findings/results and future study in this session.

 Response: This Conclusions section was rewrote as follows: “(1) Impact factors of HLDs included slope, elevation, slope aspect, lithology, distance to faults, distance to rivers, NDVI, land use, mean precipitation, profile curvature, SPI and TWI. The HLDs inventory containing 1543 disaster points and 1543 non-disaster points along 9 expressways, 15 national highways and 8 provincial highways in 15 provinces was compiled. PCA method was implemented to extract the susceptibility evaluation indexes and four principal components were obtained, whose accumulative contribution rate was 92.050%. The PSO-SVM model and GA-SVM model were used to susceptibility evaluation of HLDs in China respectively, the evaluation results of the PSO-SVM model were better than those of the GA-SVM model. Micro dangerous areas, moderate dangerous areas, severe dangerous areas and extreme dangerous areas accounted for 24.24%, 19.49%, 36.53% and 19.74% of the total areas of China, among the 1543 disaster points in the HLDs inventory, there were 134, 182, 421 and 806 located in the above areas respectively.

 (2) This study can be improved from several aspects as below: (1) The evaluation results of the PSO-SVM model are better than those of the GA-SVM model, but the AUC value was only 0.907 and the evaluation accuracy could be further improved. In addition, other evaluation methods such as the LR, ANN and information value method were not implemented and their evaluation accuracies were not verified; (2) The occurring probabilities of HLDs were considered as the dominant index of susceptibility zoning and the zoning boundaries were determined upon isometric principle, which decreased the accuracies of the susceptibility zoning results to some extent. Studies that determines the susceptibility zoning boundaries based on the cluster analysis has not been developed.”

 (12) DOI of the references should be added (you can use “https://crossref.org/").

 Response: DOIs of the references were added in the manuscript.

Response to Reviewer #3: 

 (1) The number of observation data is not sufficient for such a big study area. there should be some justification for the observed landslides and simulated susceptible areas.

 Response: The number of disaster points and non-disaster points were added to 1543 in this manuscript respectively, and a HLDs inventory containing these points were compiled according to your advice. The simulated susceptible areas were also modified because of the changes of origin data. 

 (2) Observed landslide should be ranked based on the magnitude in terms of area, damages or the volume of mass movement to be use for assessment accuracy.

 Response: A landslide inventory containing 1543 disaster points and 1543 non-disaster points along 9 expressways, 15 national highways and 8 provincial highways in 15 provinces was compiled in this study. It was difficult to present the detailed locations, areas, damages and volumes of these points in the manuscript, so I added Table 1 and Figure 2 as follows, where Table 1 showed an overview of the highway segments in the HLDs inventory, Figure 2 showed some representative disaster points. Besides, the basic occurring regulations of HLDs can be summarized as below: (1) HLDs generally occur on slopes exceed 25°, the time of occurrence is approximately 2 hours after the start of rainfall to 5 days after the end of rainfall. The mean precipitation in disaster concentration areas generally exceeds 900mm and the annual average rainstorm days exceed 6; (2) The totally volume of the 1543 HLDs is about 8.3×106 m3, differences in scales of HLDs are large, ranging from 12 m3 to 9.6×104 m3. The lithology that easily results in HLDs include silt, loess, clastic rock, mud rock, soft and flake metamorphic rock, shale, slate, soft stratum, argillization stratum and tectonically fractured stratum; (3) Earthquakes result in loosening of the mountains and provide massive loose deposits, so HLDs in the Wenchuan, Yushu and Ya’an earthquake areas are relatively more serious and the densities and scales of HLDs have significant positive correlations with the earthquake intensities.

Table 1 Highway segments in the HLDs inventory

Highway segments Quantities of disaster points and non-disaster points Highway segments Quantities of disaster points and non-disaster points

Shenda expressway (Liaoning) 57/52 G207 Baotou segment (Inner Mongolia) 36/60

Wukui expressway (Xinjiang) 49/47 G210 Yulin segment (Shaanxi) 42/58

Binbo expressway (Shandong) 44/41 G210 Dazhou segment (Sichuang) 55/45

Xihan expressway (Shaanxi) 38/60 G213 Wenchuan segment (Sichuan) 62/61

Yonglan expressway (Hunan) 61/37 G219 Pishan segment (Xinjiang) 34/37

Chengya expressway (Sichuang) 37/39 G310 Shangluo segment (Shaanxi) 39/41

Zhangwu expressway (Fujian) 41/45 G321 Mianyang segment (Sichuang) 42/46

Duzhi expressway (Guizhou) 54/51 G338 Hanzhong segment (Shaanxi) 47/54

Kaihe expressway (Yunnan) 39/39 Beijing S109 Mentougou segment 47/57

G104 Sanming segment (Fujian) 44/57 Liaoning S214 Tieling segment 57/60

G106 Huanggang segment (Hubei) 43/47 Shandong S236 Yiyuan segment 64/57

G108 Taiyuan segment (Shanxi) 38/34 Shaanxi S206 Jingbian segment 37/41

G108 Hanzhong segment (Shaanxi) 64/64 Shaanxi S302 Yuyang segment 50/42

G110 Yinchuang segment (Ningxia) 47/37 Guizhou S312 Anshun segment 61/48

G201 Changbai segment (Jilin) 36/42 Jiangxi S102 Nanchang segment 54/47

G205 Nanping segment (Fujian) 75/43 Fujian S302 Nanping segment 49/54

 Figure 2 Representative disaster points

 (3) Base on my knowledge Topo map at scale 1:250000 is too coarse to derive slope or other topographic indices. this scale may useful for site planning like suitable area for ground water recharge, but not useful to delineate risky area for highway construction which is considered as line in spatial data modeling.

 Response: The scale of the DEM used in this manuscript was modified to 1:50000, and the resolution of the impact factors of HLDs was set to 100 m×100 m in order to run the models.

 (4) Figure 11 (Hazard rationalization map of HLDs in China) sounds like hand drowning instead of GIS-based standard map classification.

 Response: This figure was drew once a time based on GIS, as follows:

 Figure 13 Susceptibility zoning map of HLDs in China

 (5) All maps need to have some map annotations including Scale bar, North arrow, Coordinate system.

 Response: Scale bar, north arrow and coordinate system were added in the manuscript as follows:

 Figure 8 Distribution of F1

 Figure 9 Distribution of F2

 Figure 10 Distribution of F3

 Figure 11 Distribution of F4

 Figure 12 Occurring probability distribution map of HLDs in China

 Figure 13 Susceptibility zoning map of HLDs in China

---

## [Decision Letter · Decision Letter 1]

23 Jun 2020

Susceptibility mapping and zoning of highway landslide disasters in China

PONE-D-20-03927R1

Dear Dr. Che,

We’re pleased to inform you that your manuscript has been judged scientifically suitable for publication and will be formally accepted for publication once it meets all outstanding technical requirements.

Kind regards,

Claudionor Ribeiro da Silva

Academic Editor

PLOS ONE

Additional Editor Comments (optional):

Reviewers' comments:

Reviewer's Responses to Questions

**Comments to the Author**

1. If the authors have adequately addressed your comments raised in a previous round of review and you feel that this manuscript is now acceptable for publication, you may indicate that here to bypass the “Comments to the Author” section, enter your conflict of interest statement in the “Confidential to Editor” section, and submit your "Accept" recommendation.

Reviewer #1: All comments have been addressed

Reviewer #2: All comments have been addressed

Reviewer #3: All comments have been addressed

2. Is the manuscript technically sound, and do the data support the conclusions?

Reviewer #1: Yes

Reviewer #2: Yes

Reviewer #3: Yes

3. Has the statistical analysis been performed appropriately and rigorously? 

Reviewer #1: Yes

Reviewer #2: Yes

Reviewer #3: Yes

4. Have the authors made all data underlying the findings in their manuscript fully available?

Reviewer #1: Yes

Reviewer #2: Yes

Reviewer #3: Yes

5. Is the manuscript presented in an intelligible fashion and written in standard English?

Reviewer #1: Yes

Reviewer #2: Yes

Reviewer #3: Yes

6. Review Comments to the Author

Reviewer #1: Excellent! Since the authors have made significant revisions according to the comments raised by all reviewers, I am supportive of this study for publication in PONE.

Reviewer #2: The authors responded well against all the reviewer comments.

Reviewer #3: All comments have been addressed and revised according to give comments in the original manuscript.

7. PLOS authors have the option to publish the peer review history of their article (what does this mean?). If published, this will include your full peer review and any attached files.

Reviewer #1: No

Reviewer #2: Yes: Afaq Ahmad

Reviewer #3: No

---

## [Editor Report · Acceptance letter]

25 Aug 2020

PONE-D-20-03927R1 

Susceptibility mapping and zoning of highway landslide disasters in China 

Dear Dr. Che:

I'm pleased to inform you that your manuscript has been deemed suitable for publication in PLOS ONE. Congratulations! Your manuscript is now with our production department. 

Kind regards, 

on behalf of

Dr. Claudionor Ribeiro da Silva 

Academic Editor

PLOS ONE